# Immunological Characterization of Proteins Expressed by Genes Located in *Mycobacterium tuberculosis*-Specific Genomic Regions Encoding the ESAT6-like Proteins

**DOI:** 10.3390/vaccines9010027

**Published:** 2021-01-07

**Authors:** Abu Salim Mustafa

**Affiliations:** Department of Microbiology, Faculty of Medicine, Kuwait University, Safat 13110, Kuwait; abu.mustafa@ku.edu.kw; Tel.: +965-2463-6505

**Keywords:** *M. tuberculosis*, RD genomic segments, ESAT6-like proteins, diagnosis, vaccine

## Abstract

The 6 kDa early secreted antigen target (ESAT6) is a low molecular weight and highly immunogenic protein of *Mycobacterium tuberculosis* with relevance in the diagnosis of tuberculosis and subunit vaccine development. The gene encoding the ESAT6 protein is located in the *M. tuberculosis*-specific genomic region known as the region of difference (RD)1. There are 11 *M. tuberculosis*-specific RDs absent in all of the vaccine strains of BCG, and three of them (RD1, RD7, and RD9) encode immunodominant proteins. Each of these RDs has genes for a pair of ESAT6-like proteins. The immunological characterizations of all the possible proteins encoded by genes in RD1, RD7 and RD9 have shown that, besides ESAT-6 like proteins, several other proteins are major antigens useful for the development of subunit vaccines to substitute or supplement BCG. Furthermore, some of these proteins may replace the purified protein derivative of *M. tuberculosis* in the specific diagnosis of tuberculosis by using interferon-gamma release assays and/or tuberculin-type skin tests. At least three subunit vaccine candidates containing ESAT6-like proteins as antigen components of multimeric proteins have shown efficacy in phase 1 and phase II clinical trials in humans.

## 1. Introduction

Using the Bacillus Calmette Guerin (BCG) vaccine against tuberculosis (TB) has shown variable protective efficacy in different parts of the world [1,2,3,4,5]. In particular, BCG vaccination does not protect against pulmonary tuberculosis in adults [6,7,8,9,10,11], which is the major manifestation of the disease in humans as 85% of TB patients have pulmonary disease [12,13,14]. Since BCG has antigens cross-reactive with *Mycobacterium tuberculosis* and the non-tuberculous environmental mycobacteria, the low efficacy or the lack of protection by BCG vaccination is suggested to be due to masking and/or blocking effects [15,16,17,18]. According to the masking hypothesis, an early sensitization with environmental mycobacteria provides some degree of protection against TB that masks the effect of the BCG vaccine due to the presence of cross-reactive antigens [19,20]. Furthermore, the immune responses induced by cross-reactive antigens due to exposure to environmental mycobacteria lead to an early clearance of antigens in the BCG vaccine, which prevents an effective immune response from being generated, rendering it a failure, and hence causing a blocking effect [21,22,23,24]. The use of *M. tuberculosis*-specific antigens as subunit vaccines is expected to overcome the problems of blocking or masking effects [25,26,27,28].

BCG vaccination also faces a problem in the diagnosis of TB by using the widely used antigenic preparation of *M. tuberculosis* known as the purified protein derivative (PPD), in the tuberculin skin test, due to the presence of the cross-reactive antigens [28,29,30,31,32,33,34]. It is expected that *M. tuberculosis*-specific antigens may overcome the problem of diagnostic inaccuracy associated with the use of PPD in BCG-vaccinated people [35,36,37,38,39,40]. Hence, studies have been conducted to identify *M. tuberculosis*-specific antigens with vaccine and diagnostic potentials [41,42,43,44,45,46,47].

An *M. tuberculosis*-specific antigen was identified for the first time from the short-term culture filtrates of *M. tuberculosis* and was designated as the early secreted antigenic target of molecular mass 6 kDa (ESAT6) [48]. Immunological studies with ESAT6, biochemically purified from the short-term culture filtrates of *M. tuberculosis*, showed that it was a major antigen of *M. tuberculosis* recognized by T cells from mice infected with *M. tuberculosis* [49]. These results were further confirmed by using the recombinant antigen and synthetic peptides of ESAT6 [50,51,52,53,54,55,56]. ESAT6 also had epitopes recognized by B cells and antibodies [57]. Studies using overlapping synthetic peptides covering the entire sequence of ESAT6 also identified it as a major T cell antigen [58,59,60,61]. The results further showed that ESAT6 contained multiple T cell epitopes [62,63,64], which were recognized by T cells in association with several human leukocyte antigen (HLA) class II molecules that are frequently expressed in humans living in different countries and geographical locations [65,66]. These results suggested that ESAT6 could be a universally useful antigen in a subunit vaccine development and/or in diagnostic applications, and its use will not be limited due to variations in the expression of HLA molecules in different population groups [65]. Further studies confirmed the potential of ESAT6 for the specific diagnosis of TB [67,68,69,70], and its potential in the development of new subunit vaccines, either alone or in combination with other cross-reactive antigens [71,72,73,74,75].

To identify additional *M. tuberculosis* antigens and genomic regions, a subtractive genome hybridization approach was used by Mahairas et al. [76]. They identified three regions of differences (RDs), i.e., RD1, RD2, and RD3 between *M. tuberculosis* and BCG, and predicted genes in these regions for encoding 11 proteins from RD1, 13 proteins from RD2, and 12 proteins from RD3 [76]. Further analysis showed that RD1 and RD3 were absent in all BCG strains, whereas RD2 was absent from some BCG strains but present in others [76]. However, RD3 was also absent from most clinical isolates of *M. tuberculosis* [76]. Hence the antigens encoded by RD3 will not have any value in the vaccine or diagnostic applications. Among the proteins encoded by RD2, MPT64 (Rv1980c) has been identified as a dominant antigen having multiple epitopes and being presented to T cells in association with several HLA class II molecules [77,78,79,80,81]. Furthermore, MPT64 has been shown to have vaccine potential in animals either alone [82], or in combination with other *M. tuberculosis* antigens [83,84,85,86], However, MPT64 may not be useful in the specific diagnosis of TB because several BCG strains used for vaccination of people in different parts of the world express this antigen [87,88].

The proteins encoded by RD1 are considered more promising in vaccine applications because this region is present in all clinical *M. tuberculosis* isolates [89,90,91,92]. Furthermore, RD1 is absent in all sub-strains of BCG [93,94,95,96] because of its deletion during the attenuation of the parent BCG strain obtained by prolonged sub-culturing of pathogenic *M. bovis* in an artificial medium [97], and hence the application of RD1-encoded immunodominant antigens in the diagnosis of TB is not expected to have any effect due to BCG vaccination [98,99,100,101]. Interestingly, the ESAT6 gene is located in RD1, and the gene for another low molecular weight and immunodominant protein, known as the 10 kDa culture filtrate protein (CFP10), is also present in the RD1 region [102]. The genes for ESAT6 and CFP10 are located in close proximity in *M. tuberculosis* genome (Figure 1), and the two proteins are secreted as dimers [103]. Both of these proteins require the ESAT6 (ESX)-1 secretion system in order to be transported out of bacterial cells as dimers [104,105,106].

The availability of the complete genome sequence of *M. tuberculosis* [107] and its comparison with BCG identified a total of 10 RDs, other than RD1 (Table 1), which are deleted in all strains of BCG that are being used for vaccination against TB in different parts of the world [108,109].

The experiments with the peptide pools of proteins encoded by all *M. tuberculosis*-specific RDs for determination of immunological reactivity showed that RD1, RD7, and RD9 contained the immunodominant antigens recognized by T cells from TB patients [110,111,112,113,114,115]. Further analysis of individual proteins encoded by genes present in RD1, RD7, and RD9 identified several major protein antigens of *M. tuberculosis* [116,117,118,119]. In this review, all the individual proteins encoded by the genes present in RD1, RD7, and RD9 have been identified and analyzed for their putative roles, including their immunological applications in the diagnosis of TB and vaccine developments.

## 2. RD1 Genes and Encoded Proteins

The RD1 region contains the genes for EsxA (ESAT6) and EsxB (CFP10) along with 11 other M. tuberculosis-specific genes predicted by Robertson and Thole corresponding to the open reading frames (ORFs) known as ORF2 to ORF14 [120], seven genes predicted by Mahairas et al. and designated as ORF1A to ORF1K [76], and nine genes predicted by Cole et al. and designated as Rv3871 to Rv3879 in *M. tuberculosis* H37Rv genome [107] (Table 2).

The analysis of ORF genes for expression at mRNA and/or protein levels showed that at least 12/13 of them were expressed in *M. tuberculosis* [121,122,123,124,125,126]. In addition to ESAT6 (EsxA) and CFP10 (EsxB), Rv3871, PE35, PPE68, Rv3878, and Rv3879c have also been identified as major T cell antigens [127,128,129,130,131,132,133,134,135,136,137,138,139,140]. Concerning the functions, EsxA and EsxB are associated with deactivation of macrophage and dendritic cell functions and are involved in the virulence of *M. tuberculosis* [141,142,143,144,145]. Among the RD1 proteins, Rv3871, Rv3872 (PE35), Rv3873 (PPE68), ESAT6 (EsxA), CFP10 (EsxB) and Rv3879c have been suggested for use in the diagnosis of TB using T cell assays [146,147,148,149]. In fact, commercial tests developed to diagnose TB using interferon-gamma release assays include a cocktail of antigens including ESAT6 and CFP10 [150,151,152,153,154]. Furthermore, several RD1 antigens, i.e., ESAT6, CFP10, Rv3871, Rv3872 (PE35), Rv3873 (PPE68), Rv3872, Rv3876 Rv3879, and ORF14 have also been suggested for use in antibody assays for the specific diagnosis of TB [155,156,157,158,159,160,161,162,163]. Moreover, RD1 antigens have also been tested in tuberculin type response with encouraging results [164,165,166,167,168,169,170,171].

In addition to the role of the whole RD1 genomic segment in protective immunity [172,173], the evaluations of individual RD1 proteins for vaccine development in animals have shown the potential of PE35, PPE68, ESAT6, and CFP10 in the development of new subunit vaccines for TB [174,175,176,177,178,179,180,181]. Animal experiments with recombinant (r)BCG strains expressing RD1 antigens have shown the induction of protective type immune responses and protected the immunized animals after challenges with *M. tuberculosis* [182,183,184,185,186]. Furthermore, a rBCG vaccine candidate containing Ag85B, ESAT6, and CFP10 (GamTBvac) showed a strong protective effect as a BCG booster vaccine in mice and guinea pigs [187]. Phase 1 and Phase 2 clinical trials with GamTBvac have been conducted. In both types of clinical trials, the results after vaccination with GamTBvac showed that the vaccine had an acceptable safety profile and induced markers of protective immunity, i.e., antigen-specific interferon gamma release, Th1 cytokine-expressing CD4+ T cells, and IgG responses [188,189]. These results support further clinical evaluation of GamTBvac in Phase 3 trials to evaluate its efficacy in protecting against infection with *M. tuberculosis* and the development of the clinical disease.

Two other subunit vaccine candidates which have undergone clinical trials in humans are H1:IC31 and H56:IC31 [190,191]. The subunit vaccine H1:IC31 contains a fusion of Ag85B and ESAT6 and is given along with the adjuvant IC31 [192,193]. H56:IC31 contains the RD1 antigen ESAT6 and two other *M. tuberculosis* proteins, Ag85B and Rv2660c, as a fusion protein, and it is used for immunization along with the adjuvant IC31 [193]. Both Ag85B and Rv2660c are major antigens of *M. tuberculosis* and are cross-reactive with BCG and other mycobacteria [194,195,196]. The subunit vaccines H1:IC31 and H56:IC31 have induced protective type cellular immune responses in animals and protected them upon being challenged with the virulent *M. tuberculosis* [193,197,198].

Humans vaccinated with H1:IC31 vaccine did not show local or systemic adverse effects except transient soreness at the injection site, but there was induction of strong antigen-specific T cell responses, which persisted through 30 months of follow-up. This indicated the activation of a substantial memory response in the vaccinated subjects [199]. The H1:IC31 vaccine was also safe and well tolerated in HIV-infected adults with a CD4+ lymphocyte count greater than 350 cells/mm^3^ and induced a specific and long-lasting Th1 immune response [200]. The H1:IC31 vaccine was further tested in a phase 1, open-label trial in people living in a TB-endemic area. Healthy male participants aged 18–25 years were recruited into four groups. Participants in group 1 and group 2 were Tuberculin Skin Test (TST) negative and QuantiFERON-TB Gold in-tube test (QFT) negative (Mtb-naïve groups), participants in group 3 were TST positive and QFT negative (BCG group), and participants in group 4 were both TST and QFT positive (Mtb-infected group). The results showed that the vaccine was safe and generally well tolerated [201]. Immunogenicity assays showed a stronger response to TB antigens in the Mtb-naïve group [201]. H1:Ic31 has also undergone a phase 2 clinical trial in 240 healthy adolescents in South Africa including both *M. tuberculosis*-infected and non-infected subjects [202]. No noticeable safety events were observed in any group irrespective of the doses or vaccination schedule used [202]. Furthermore, the vaccine induced antigen-specific CD4+ T cell responses of protective phenotype in both the groups [202].

A double blind, placebo-controlled, dose selection trail in humans for dose optimization of H56:Ic31 in a tuberculosis-endemic population showed that two or three vaccinations at the lowest dose induced long-lasting antigen-specific CD4 T cell responses with acceptable safety profiles in both naïve and *M. tuberculosis*-infected subjects [203]. A phase 1b randomized study with the H56:IC31 vaccine showed that the vaccine had acceptable safety profiles in *M. tuberculosis*-uninfected adults and induced immunizing antigen-specific cellular and humoral immune responses [204]. This vaccine candidate is now being tested in phase 2a clinical trials, and recruitment has started for the phase 2b clinical trial [190].

## 3. RD7 Genes and Encoded Proteins

The *M. tuberculosis*-specific genomic region RD7 contains eight ORFs, and an equal number of genes are annotated in the *M. tuberculosis* H37Rv genome [205] (Table 3).

In human studies with TB patients, the mixture of peptides corresponding to all eight proteins showed that RD7 contains immunodominant antigens stimulating the immune response with a Th1-bias [110,111,114]. Two of the proteins encoded by genes in RD7 belong to the ESAT6 family and are known as EsxO (Rv2346c) and EsxP (Rv2347c) (Table 3). In tuberculin positive reactor cattle, EsxO (Rv2346c) and EsxP (Rv2347c) induced significant IFN-γ responses in vitro [206]. Further experiments with respect to the diagnostic value of ESAT6-like proteins showed that 57% TB reactor cattle responded to EsxO (Rv2346c) peptides in IFN-γ assays, without inducing positive responses in any of the BCG-vaccinated animals [207]. By using human T cell clones and a synthetic peptide library consisting of 15-mers overlapping by 11 aa, Lewinsohn et al. have shown that Rv2347c is an antigen capable of stimulating IFN-γ secreting CD8^+^ T cells [208]. In mice, immunizations with EsxO (Rv2346c) and EsxP (Rv2347c) using different delivery systems, i.e., chemical adjuvants, mycobacteria and naked plasmids, showed that both of these antigens induced protective Th1 responses but none of them induced pathologic Th2, Th17 and T regulatory cell responses [209]. However, none of these antigens have been tested in the diagnosis of TB in humans or in vaccine development.

## 4. RD9 Genes and Encoded Proteins

The *M. tuberculosis*-specific genomic region of RD9 contains seven ORFs and all of them are annotated as genes in the *M. tuberculosis* H37Rv genome [205] (Table 4).

The immunological evaluation of RD9 proteins using synthetic peptides showed that this region also encodes immunodominant proteins [110,111,114]. Among the RD9 proteins are included two ESAT6 family proteins, i.e., Rv3619c (EsxV) and Rv3620c (EsxW) (Table 4). Molecular modeling and docking studies predicted that the structure of Rv3619c-Rv3620c was similar to that of ESAT6-CFP10 [210]. Immunization with Rv3619c and/or Rv3620c proteins, either alone or in combination with other *M. tuberculosis* proteins, induced antigen-specific humoral and cellular immune responses in mice [174,175,210,211]. Immunizations of mice with Rv3619c protected against a challenge with *M. tuberculosis* [212], and allergic asthma [213].

A multiprotein vaccine candidate, ID93, containing Rv2608, Rv3619c, Rv3620c, and Rv1813 *M. tuberculosis* antigens, combined with synthetic toll-like receptor 4 (TLR4) agonist glucopyranosyl lipid adjuvant (GLA) in a stable nano-emulsion (SE) has been developed and is known as ID93/GLA-SE [214]. Immunization of mice with ID93/GLA-SE did not induce sensitivity to the proteins present in PPD, hence it may not compromise the diagnostic efficacy of PPD in the diagnosis of TB [215]. In contrast, positive delayed-type hypersensitivity reactions to ID93 and its components were induced in ID93/GLA-SE-immunized animals, which indicated the induction of strong but specific cellular immune responses in the immunized animals [215].

Furthermore, immunizations with ID93/GLA-SE protected animals upon challenge with a clinical isolate of *M. tuberculosis* as well as the hyper-virulent Korean Beijing strain K of *M. tuberculosis* and induced long-lived immunity in mice [216,217]. Moreover, therapeutic immunizations with ID93/GLA-SE induced differential T cell immune responses over the course of infection that correlated with periods of enhanced bacterial control over that of drug treatment alone in mice [218]. In a BCG-prime boost regimen, the ID93/GLA-SE vaccine significantly reduced bacterial load at 16 weeks after challenge with the hyper-virulent Beijing strain of *M. tuberculosis*, while the BCG vaccine alone did not confer significant protection [219].

A randomized, double-blind, placebo-controlled phase 1 clinical trial with the ID93/GLA-SE vaccine has been conducted in HIV-negative and previously BCG-vaccinated adults in South Africa. The participants included *M. tuberculosis* infected and non-infected healthy subjects. The results using varying doses of the vaccine showed that it was well-tolerated and no severe or serious vaccine-related adverse events were observed [220]. Furthermore, different doses of the vaccine did not affect the frequency or severity of adverse events, but mild injection site adverse events and flu-like symptoms were common in *M tuberculosis*-infected group compared to non-infected group. Vaccination induced long-lasting antigen-specific IgG and T helper-1 type cellular immune responses, which peaked after administration of two doses of the vaccine [220]. The variations in the vaccine dose did not significantly affect the magnitude, kinetics, or profile of antibody and cellular immune responses [220]. When compared with vaccination with ID93 alone, vaccination with ID93 + GLA-SE induced higher titers of ID93-specific antibodies, a preferential increase in IgG1 and IgG3 antibody subclasses, and a multifaceted Fc-mediated effector function response [220]. The ID93/GLA-SE vaccine enhanced the magnitude and polyfunctional cytokine profile of CD4^+^ T cells, as compared to ID93 alone [221]. This vaccine is currently being tested in a phase 2 clinical trial [190].

## 5. Sequence Identities among ESAT6-Like Proteins Encoded by Genes in RD1, RD7 and RD9

All of the six ESAT6-like proteins encoded by genes in RD1, RD7 and RD9 are of low molecular weight, are approximately the same size and share a similar genomic organization [102], but they share minimum sequence identities (6 to 20%) with ESAT6 (EsxA) (Table 5) and CFP10 (EsxB) [102,168], suggesting that none of these proteins can replace ESAT6 (EsxA) and/or CFP10 (EsxB) either in diagnosis or vaccine applications.

Similarly, the individual ESAT6-like proteins encoded by genes in RD7 and RD9 also have minimum sequence identities with each other (Table 6). In contrast, the EsxV (Rv3619c) and EsxW (Rv3620c) encoded by RD7 have extensive sequence identities (93% and 97%) with EsxO (Rv2346c) and EsxP (Rv2347c) encoded by RD9, respectively (Table 6). This suggests that EsxV (Rv3619c) and EsxO (Rv2346c); and EsxW (Rv3620c) and EsxP (Rv2347c) have evolved through gene duplication. These high levels of sequence identities suggest that EsxV (Rv3619c) may replace EsxO (Rv2346c) and EsxW (Rv3620c) may replace EsxP (Rv2347c) in diagnostic and/or vaccine applications.

## 6. Summary

The *M. tuberculosis*-specific genomic regions RD1, RD7, and RD9 can potentially encode a total of 29 proteins. Among them, three pairs of proteins, i.e., EsxA (ESAT6) and EsxB (CFP10); EsxO (Rv2346c) and EsxP (Rv2347c); and EsxV (Rv3619c) and EsxW (3620c) belong to the family of ESAT6-like proteins. EsxO and EsxV share 93%, and EsxP and EsxW share 97% sequence identities, suggesting gene duplication. EsxA and EsxB have been widely used in the specific diagnosis of infection with *M. tuberculosis* in interferon-gamma release assays. Furthermore, both of these proteins have also been included as components of subunit vaccines that have been tested in human phase 1 and phase 2 clinical trials. However, EsxA and EsxB cannot be used both as vaccines and diagnostic reagents. Furthermore, EsxV and EsxW are also included in a vaccine preparation known as ID93, which is undergoing a phase 2a clinical trial in humans. Due to the extensive use of EsxA and EsxB in diagnostic applications, it is advisable to exclude them from vaccine preparations and focus on the use of the vaccine candidates containing EsxV and EsxW, i.e., ID93 for vaccination against TB.

## Figures and Tables

**Figure 1 vaccines-09-00027-f001:**
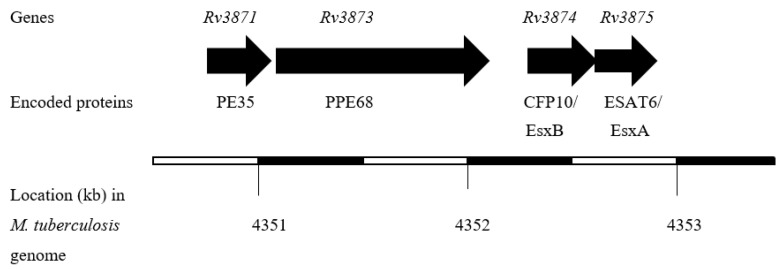
The organization of the CFP10 and ESAT6 genes in *M. tuberculosis* genome.

**Table 1 vaccines-09-00027-t001:** *M. tuberculosis* genomic regions deleted in all strains of Bacillus Calmette Guerin (BCG) and annotations of deleted genes in the lab-adopted virulent *M. tuberculosis* strain H37Rv.

Region Deleted (RD) ^a^	Annotations (Rv nos.) of Deleted Genes ^b^
RD1	Rv3871-Rv3879c
RD4	Rv0221-Rv0223c
RD5	Rv3117-Rv3121
RD6	Rv1506c-Rv1516c
RD7	Rv2346c-Rv2353c
RD9	Rv3617-Rv3623
RD10	Rv1255c-Rv1257c
RD11	Rv3425-Rv3429
RD12	Rv2072c-Rv2075c
RD13	Rv2645–Rv2660c
RD15	Rv1963c-Rv1977

^a^ Regions Deleted (RDs) are numbered according to Behr et al. [108]. ^b^ Rv nos. of deleted genes are assigned in *M. tuberculosis* strain H37Rv genome according to Behr et al. [108].

**Table 2 vaccines-09-00027-t002:** Gene annotation, gene name, and description of proteins encoded by genes in RD1.

Amoudy et al. [120]	Mahairas et al. [76]	Cole et al. [107]	Gene Name	Description of Proteins
ORF2	ORF1A	Rv3871	*Rv3871*	591 aa, Probable conserved hypothetical protein
ORF3	NP	Rv3872	*PE35*	99 aa, PE family-related protein
ORF4	NP	NP	*orf4*	139 aa, Hypothetical protein
ORF5	ORF1B	Rv3873	*PPE68*	368 aa, PPE family protein
ORF6	NP	Rv3874	*esxB*	100 aa, 10 kDa culture filtrate antigen EsxB (LHP, CFP10, MTSA10)
ORF7	ORF1C	Rv3875	*esxA*	95 aa, 6 kDa early secretory antigenic target EsxA (ESAT6)
ORF8	NP	NP	*orf8*	140 aa, Hypothetical protein
ORF9	ORF1D	Rv3876	*Rv3876*	668 aa, Conserved hypothetical proline and alanine rich protein
ORF10	ORF1E	Rv3877	*Rv3877*	511 aa, Probable conserved transmembrane protein
ORF11	ORF1F	Rv3878	*Rv3878*	281 aa, Conserved hypothetical alanine-rich protein
ORF12	ORF1G	none	*orf12*	564 aa, Hypothetical protein
ORF13	ORF1K	Rv3879	*Rv38789*	729 aa, Hypothetical alanine and proline rich protein
ORF14	NP	NP	*orf14*	263 aa, Hypothetical protein, recognized by antibodies present in sera of TB patients
ORF15	NP	NP	*orf15*	96 aa, Hypothetical protein

NP = Not predicted.

**Table 3 vaccines-09-00027-t003:** Open reading frame (ORF) code, gene annotation, gene name and description of proteins encoded by genes in RD7.

ORF Code	Rv Gene Annotation	Gene Name	Description of Proteins
ORF1	Rv2346c	*EsxO*	94 aa, ESXO, MTB9.9E, ESAT6-like protein 6
ORF2	Rv2347c	*EsxP*	98 aa, ESXP, QILSS, ESAT6-like protein 7
ORF3	Rv2348c	*Rv2348c*	108 aa, Rv2348c, hypothetical unknown protein
ORF4	Rv2349c	*plcC*	508 aa, PLCC, probable phospholipase C 3
ORF5	Rv2350c	*plcB*	512 aa, PLCB, probable membrane-associated phospholipase C 2
ORF6	Rv2351c	*plcA*	512 aa, PPLCA, MTP40 antigen, probable membrane-associated phospholipase C 1
ORF7	Rv2352c	*PPE38*	392 aa, PPE38, PPE family protein
ORF8	Rv2353c	*PPE39*	354 aa, PPE39, PPE family protein

**Table 4 vaccines-09-00027-t004:** ORF code, gene annotation, gene name and description of proteins encoded by genes in RD9.

ORF Code	Rv Gene Annotation	Gene Name	Description of Proteins
ORF1	Rv3617	*ephA*	322 aa, EPHA, probable epoxide hydrolase, (Epoxide hydratase) (Arene-oxide hydratase)
ORF2	Rv3618	*Rv3618*	395 aa, Rv3618, possible monooxygenase
ORF3	Rv3619c	*esxV*	94 aa, EsxV, ESAT6 family protein
ORF4	Rv3620c	*esxW*	98 aa, EsxW, ESAT6 family protein
ORF5	Rv3621c	*PPE65*	414 aa, PPE65, PPE-family protein
ORF6	Rv3622c	*PE32*	99 aa, PE32, PE family protein
ORF7	Rv3623	*lpqG*	240 aa, LPQG, probable conserved lipoprotein

**Table 5 vaccines-09-00027-t005:** The amino acid sequence identities between ESAT6, and ESAT6-like proteins encoded by RD1, RD7 and RD9.

Protein	Comparison with	Protein Identity
ESAT6 (EsxA)	CFP10 (EsxB)	15%
ESAT6 (EsxA)	EsxV (Rv3619c)	20%
ESAT6 (EsxA)	EsxW (Rv3620c)	6%
ESAT6 (EsxA)	EsxO (Rv2346c)	18%
ESAT6 (EsxA)	EsxP (Rv2347c)	6%

**Table 6 vaccines-09-00027-t006:** The amino acid sequence identities between ESAT6-like proteins encoded by RD7 and RD9.

Protein	Comparison with	Percent Identity
EsxV (Rv3619c)	EsxW (Rv3620c)	30%
EsxO (Rv2346c)	EsxP (Rv2347c)	<5%
EsxV (Rv3619c)	EsxO (Rv2346c)	93%
EsxW (Rv3620c)	EsxP (Rv2347c)	97%

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
