# Peer review of "Immunological Characterization of Proteins Expressed by Genes Located in Mycobacterium tuberculosis-Specific Genomic Regions Encoding the ESAT6-like Proteins"

_vaccines, 2021, doi:10.3390/vaccines9010027_

Round 1

Reviewer 1 Report

The paper is focused on RD regions of the MTB genome and their involvement in producing an immune response.  It particularly emphasizes how these regions encode for proteins similar to ESAT6, an already known antigen, and their immunogenic potential. The writing style is satifactory and the flow is inconsistent in places. 

The major problem with the review is:

  1. Lack of a graphic portraying how these RDs differ and what is unique about each one of them. 
  2. lack of original ideas that better build a case for one or the other ESX proteins. 
  3. The discussion section does try to build a case for other ESX proteins but doesn't elaborate much on their potential, challenges and possible ways to address these challenges when it comes to using these alternative antigens. It's information/ideas like such that contribute towards citations as this reflects the originality of thought. For example, the author completely ignores the point that for EsxO and EsxP to replace the EsxV and EsxW they're supposed to ilicit an immune response. This caveat need be discussed. A similarity in sequence identity is a good indicator that this could be the case but one should refrain from offering the two as an outright substitution candidate. 

I urge the author to better develop the discussion section on above mentioed lines to have a better citation index w.r.t. this review paper. 

Minor points include:

  1. L12  The gene encoding for ESAT6 instead of The gene for ESAT6
  2. Lines 34-37 makes no sense and should be re-drafted to make it clear what the author is trying to imply. My best guess is that the author wants to make a case for the early clearance of antigens in the BCG vaccine which prevents an immune response to be generated rendering it a failure, which is officaly termed as the blocking/masking effect?  Please re-write these lines. 
  3. Author may consider if Lines 34-37 read better after Lines 39- 44 as they all discuss cross-reactivity. This would probably make for a better flow. 
  4. L168, L174 state the antigen to be H1:Ic31 vs L176 as H56:IC31 please correct. 
  5. L202 please change "and also protected against allergic asthma" to "allergic asthma". 
  6. L207 "hence they will not compromise" is a bold claim. Change it to "hence may not" 
  7. L228-229 "data not shown" doesn't work usually in a review. Please add a rationale expanding on this claim.

Author Response

Point 1: Lack of a graphic portraying how these RDs differ and what is unique about each one of them.

Response 1: The author would like to thank the reviewer for this comment but would like to differ from his/her opinion of graphical portraying. I think the tables provide much better information than graphical presentation of the information provided. The second reviewer agrees to my point in his remarks, as given below.

“This research addresses the need for a comprehensive list of proteins encoded by RD1, RD7 and RD9 region genes of M. tuberculosis. For example, the table 2 summarizes the gene annotation, gene name, and description of proteins encoded by genes in RD1, while the table 3 and 4 show the gene annotation, gene name, and description of proteins encoded by genes in RD7 and RD9 respectively. Having such a research where the RDs regions involved in the virulence of Mycobacterium tuberculosis are studied represents a great resource for investigators in the field especially for early career scientists. The tables are detailed and well summarized. In every section, the author made an analysis and comment on the immunological characterization of proteins expressed by the RD1, 7 and 9 genes.”

Point 2: lack of original ideas that better build a case for one or the other ESX proteins.

Response 2: The idea of this review is not to build original ideas that better build a case for one or the other ESX proteins. The idea is to review the information available in the literature with respect to the diagnostic and vaccine applications of ESX proteins.

Point 3: The discussion section does try to build a case for other ESX proteins but doesn't elaborate much on their potential, challenges and possible ways to address these challenges when it comes to using these alternative antigens. It's information/ideas like such that contribute towards citations as this reflects the originality of thought. For example, the author completely ignores the point that for EsxO and EsxP to replace the EsxV and EsxW they're supposed to illicit an immune response. This caveat need to be discussed. A similarity in sequence identity is a good indicator that this could be the case but one should refrain from offering the two as an outright substitution candidate. 

I urge the author to better develop the discussion section on above mentioned lines to have a better citation index w.r.t. this review paper. 

Response 3:

This is a review article and not an original article. In authors opinion, and the opinion of the second reviewer, all relevant points were discussed in the submitted manuscript. However, some relevant additions have been made in the revised manuscript.

Minor points include:

Point 4: L12  The gene encoding for ESAT6 instead of The gene for ESAT6

Response 4: It has been changed to “The gene encoding ESAT6” in the revised manuscript.

Point 5: Lines 34-37 makes no sense and should be re-drafted to make it clear what the author is trying to imply. My best guess is that the author wants to make a case for the early clearance of antigens in the BCG vaccine which prevents an immune response to be generated rendering it a failure, which is officially termed as the blocking/masking effect?  Please re-write these lines.

Response 5: The sentence has been modified in the revised manuscript, as given below.

Furthermore, the immune responses induced to cross-reactive antigens due to exposure to environmental mycobacteria lead to an early clearance of antigens in the BCG vaccine, which prevents an effective immune response to be generated, rendering it a failure, and hence a blocking effect ]4].

Point 6: Author may consider if Lines 34-37 read better after Lines 39- 44 as they all discuss cross-reactivity. This would probably make for a better flow. 

Response 6: The author considers that the lines 34-37 remain where they are because these lines and the paragraph mention the shortcomings of BCG as a vaccine and the lines 39-44 are the next paragraph related to the problem of BCG vaccination in application of PPD in the diagnosis of TB.

Point 7: L168, L174 state the antigen to be H1:Ic31 vs L176 as H56:IC31 please correct. 

Response 7: I would request the referee to recheck the previously submitted manuscript. Only L168 stated the antigen to be H1:Ic31, whereas L174 and L176 stated the antigen to be H56:IC31. The statements were supported by the appropriate references in the submitted manuscript, i.e. Reference 135 (131 in the revised manuscript) for H1:Ic3, and 136,137 (132, 133 in the revised manuscript) for H56:Ic31.

Point 8: L202 please change "and also protected against allergic asthma" to "allergic asthma". 

Response 8: It has been changed to “allergic asthma” in the revised manuscript

Point 9: L207 "hence they will not compromise" is a bold claim. Change it to "hence may not" 

Response 9: "hence they will not compromise" has been changed to "hence it may not compromise" in the revised manuscript.

Point 10: L228-229 "data not shown" doesn't work usually in a review. Please add a rationale expanding on this claim.

Response 10: “Data not shown” has been deleted and a relevant reference [ no. 100] has been added in the revised manuscript.

Reviewer 2 Report

In this review untitled: “Immunological characterization of proteins expressed by genes located in Mycobacterium tuberculosis specific genomic regions encoding the ESAT6-like proteins”, the author reviews the literature of three Mycobacterium tuberculosis region of differences (RD1, RD7 and RD9) to analyze the putative role of these RDs and their immunological applications in the diagnosis of TB and vaccine developments. The focus on the three RDs was justified by the fact that they are the main RDs which contain the immunodominant antigens recognized by T cells from TB patients. This is a well written and “straight forward to the point” review and below are some few minor comments.

  1. Line 160: well tolerated in in HIV-infected adults.... remove “in”
  2. Line 225: ... 5. Sequence identities among ESAT6-like proteins encoded by genes in RD1, RD7 and RD9: ... remove “:”?
  3. Line 228: “data not shown”. Since this is a review, I suggest that the author refer to the paper as reference instead of mentioning “data not shown”.

Additional comments:

This research addresses the need for a comprehensive list of proteins encoded by RD1, RD7 and RD9 region genes of M. tuberculosis. For example, the table 2 summarizes the gene annotation, gene name, and description of proteins encoded by genes in RD1, while the table 3 and 4 show the gene annotation, gene name, and description of proteins encoded by genes in RD7 and RD9 respectively. Having such a research where the RDs regions involved in the virulence of Mycobacterium tuberculosis are studied represents a great resource for investigators in the field especially for early career scientists. The tables are detailed and well summarized. In every section, the author made an analysis and comment on the immunological characterization of proteins expressed by the RD1, 7 and 9 genes. The author previously published a paper entitled “Identification of Mycobacterium tuberculosis-specific genomic regions encoding antigens inducing protective cellular immune responses” in the Indian J Exp Biol in 2009 where he suggests the relevance of RD1-encoded proteins in pathogenesis of tuberculosis. An overall understanding of the immunological action of RD regions involved in the virulence of M. tuberculosis will be useful for designing and developing more effective TB vaccines which are currently not so effective when referred to BCG.

Regarding the original of the topic , on a scall of 1 to 5, the topic is original at 4. I found the focus on RD7 and RD9 to add more value to this review as most of the published papers and reviews focused on RD1 antigens for instance.

The paper is well written and the text in clear and easy to read.
I would recommend to the author to improve the section “3. RD7 genes and encoded proteins” which I think is a little short.

Author Response

Point 1: In this review untitled: “Immunological characterization of proteins expressed by genes located in Mycobacterium tuberculosis specific genomic regions encoding the ESAT6-like proteins”, the author reviews the literature of three Mycobacterium tuberculosis region of differences (RD1, RD7 and RD9) to analyze the putative role of these RDs and their immunological applications in the diagnosis of TB and vaccine developments. The focus on the three RDs was justified by the fact that they are the main RDs which contain the immunodominant antigens recognized by T cells from TB patients. This is a well written and “straight forward to the point” review and below are some few minor comments.

Response: The author thanks the reviewer for stating that “this is a well written and straight forward to the point review”. No comments to respond.

Point 1. Line 160: well tolerated in in HIV-infected adults.... remove “in”

Response 1: The word “in” has been removed in the revised manuscript.

Point 2: Line 225: ... 5. Sequence identities among ESAT6-like proteins encoded by genes in RD1, RD7 and RD9: ... remove “:”?

Response 2: The author would like to keep this title because the following text compares the sequences of ESAT6-like proteins encoded by genes in RD1, RD7 and RD9.

Point 2: Line 228: “data not shown”. Since this is a review, I suggest that the author refer to the paper as reference instead of mentioning “data not shown”.

Response 3: The reference has been mentioned and “data not shown” has been deleted in the revised manuscript.

Additional comments:

Point 4: This research addresses the need for a comprehensive list of proteins encoded by RD1, RD7 and RD9 region genes of M. tuberculosis. For example, the table 2 summarizes the gene annotation, gene name, and description of proteins encoded by genes in RD1, while the table 3 and 4 show the gene annotation, gene name, and description of proteins encoded by genes in RD7 and RD9 respectively. Having such a research where the RDs regions involved in the virulence of Mycobacterium tuberculosis are studied represents a great resource for investigators in the field especially for early career scientists. The tables are detailed and well summarized. In every section, the author made an analysis and comment on the immunological characterization of proteins expressed by the RD1, 7 and 9 genes. The author previously published a paper entitled “Identification of Mycobacterium tuberculosis-specific genomic regions encoding antigens inducing protective cellular immune responses” in the Indian J Exp Biol in 2009 where he suggests the relevance of RD1-encoded proteins in pathogenesis of tuberculosis. An overall understanding of the immunological action of RD regions involved in the virulence of M. tuberculosis will be useful for designing and developing more effective TB vaccines which are currently not so effective when referred to BCG.

Regarding the original of the topic , on a scall of 1 to 5, the topic is original at 4. I found the focus on RD7 and RD9 to add more value to this review as most of the published papers and reviews focused on RD1 antigens for instance.

Response 4: The author thanks the reviewer for positive remarks. No comments to respond.

Point 5. The paper is well written and the text in clear and easy to read.
I would recommend to the author to improve the section “3. RD7 genes and encoded proteins” which I think is a little short.

Response 5: The author thanks the reviewer for positive remarks. As suggested by the reviewer, the section 3 has been improved and expanded (lines 194 to 206 in the revised manuscript).

Round 2

Reviewer 1 Report

None to be discussed.